# Finite Element Assessment of a Hybrid Proposal for Hip Stem, from a Standardized Base and Different Activities

Manuel Guzmán [1], Emmanuel Durazo [1,*], Alejandro Ortiz [1], Israel Sauceda [1], Miriam Siqueiros [1], Luis González [1] and David Jiménez [2]

1   Facultad de Ingeniería, Universidad Autónoma de Baja California, Mexicali 21280, Baja California, Mexico
2   Institute of Medical and Biological Engineering, School of Mechanical Engineering, University of Leeds, Leeds LS2 9JT, UK
*   Correspondence: emmanuel.durazo@uabc.edu.mx

**Abstract:** Choosing a suitable prosthesis to restore the functionality of the hip joint is a complex problem. The stem geometries, materials, and type of hip damage are critical factors for avoiding potential issues (aseptic loosening, fracture, and natural wear and tear). Comparing the available stems to select the best option is not straightforward because of the various loads and boundary conditions used in the tests, making the process difficult to compare the advantages and disadvantages among them. This work proposes stem assessment using a standardized base (generated from a literature review and ISO standards) to compare the stem geometries and present a new hybrid design to improve performance using the best qualities of the implants reported in the literature review. Sixteen hip prostheses were evaluated with the finite element method (FEM) using the same boundary and loading conditions through multi-objective analysis (von Mises stress and strain). Consequently, a hybrid geometry proposal was obtained by assessing specific points through the stem length (medial and lateral region) to define the cross-section (trapezoidal) and the new profile. The new hybrid implant proposal presented a stress reduction of 9.6% when compared to the reference implant P2-T (the implant with the best behavior) in the most critical activity (activity 4) using a titanium alloy. A similar stress reduction of 9.98% was obtained using ASTM F2996-13 and ISO 7206–4:2010(E) standards.

**Keywords:** hip joint; aseptic loosening; standardized base; finite element method; hybrid implant

## 1. Introduction

Total hip arthroplasty involves replacing the acetabular surface and the femoral head with prosthetic implants, known as total hip replacement (THR), to recover the joint functionality after suffering mechanical damage or a degenerative disease [1–4]. Prosthesis selection for THR depends on the type of fracture/pathology, the surgeon's preference, and the availability and the cost of the prostheses in the market [5–7]. There is also an increasing demand for THR in younger people, so the aim is to reach longevity of 20 to 25 years after being implanted, with an expected average lifespan of 15 years for the used stems [7,8].

The prosthesis used in THR has particular components; the main ones are the acetabular head and the stem. These components are usually made of metal alloys, despite that they are prone to loose particles, the CoCr alloys, stainless steel alloys (SS 316L), and titanium alloys (TI6Al4V) being the most frequently used [2,9–11]. Those materials have high elasticity moduli compared to the elasticity modulus of the human bone (12–30 GPa) [2,3,8,12–21], and when implanted, represent a critical factor in the stress and strain distribution. Along with the materials, improvement in the stem design is essential. A balance between the distribution of the stress and strain along the implant's length is crucial in keeping the functionality and securing the implant's bonding to the bone structure.

During stem design, a multi-objective analysis was carried out to investigate the different types of failures that can occur in a hip implant. Applications of these studies using the multi-objective analysis can be found in fracture analysis [22,23], stress shielding [24–26], micro-movements [2,27], and also in studies aiming to improve the performance of a hip implant by modifying its profile or cross-sectional area [2,9,28]. The variables typically studied in this analysis are the principal stresses, von Mises equivalent stress, total deformation, and von Mises equivalent strain. Multi-objective analysis has been used to develop and improve stem geometries, using different loads [2,9,28–37]. Then, FEM is the tool used to analyze different variables (as reported previously [38–40]), and it is regularly used to analyze the implants, allowing the evaluation of critical factors. Those analyses mainly focus on stress and strain by combining different profiles and cross-sectional areas [2,9,10,28].

We aimed to develop a new implant (hybrid implant) based on assessing sixteen implants with different profiles and cross-sections by comparing their advantages and disadvantages using the same base for the analysis. Multi-objective analysis was used to evaluate different implants. First, the cross-sectional area with the best performance was selected. Then, a new profile design was generated by analyzing various points on the medial and lateral region of the implants. Finally, combining the section area with the new profile design, a hybrid implant was developed to demonstrate that the new design has better mechanical performance than the previous implants, resulting in lower stresses and strain at the points analyzed.

## 2. Materials and Methods

Comparing the reported stems in the literature was a challenge, as they have been evaluated under different boundary conditions, materials, loads, and meshing conditions, as reported by C.K.N. et al. [41]. A standard base neededs to be stated to compare the hip implants under the same conditions. The methodology to generate the hybrid implant is presented in Figure 1, and it involved three stages:

First stage: Identification and validation process. Heuristic stem research was performed to identify its respective boundary conditions and loads. At this stage, the best option was selected, and its results were considered for validation.

Second stage: Design and analysis process. All the identified stems were parametrized and assessed using the same boundary conditions identified from the previous stage. At the same time, different loads were evaluated to determine their critical stem responses to daily activities.

Third stage: Hybrid design process. A stem design was proposed at this stage. It was based on the results obtained in stages one and two. Finally, the new stem design was analyzed and compared with the best stem results obtained from the sixteen stems analysis.

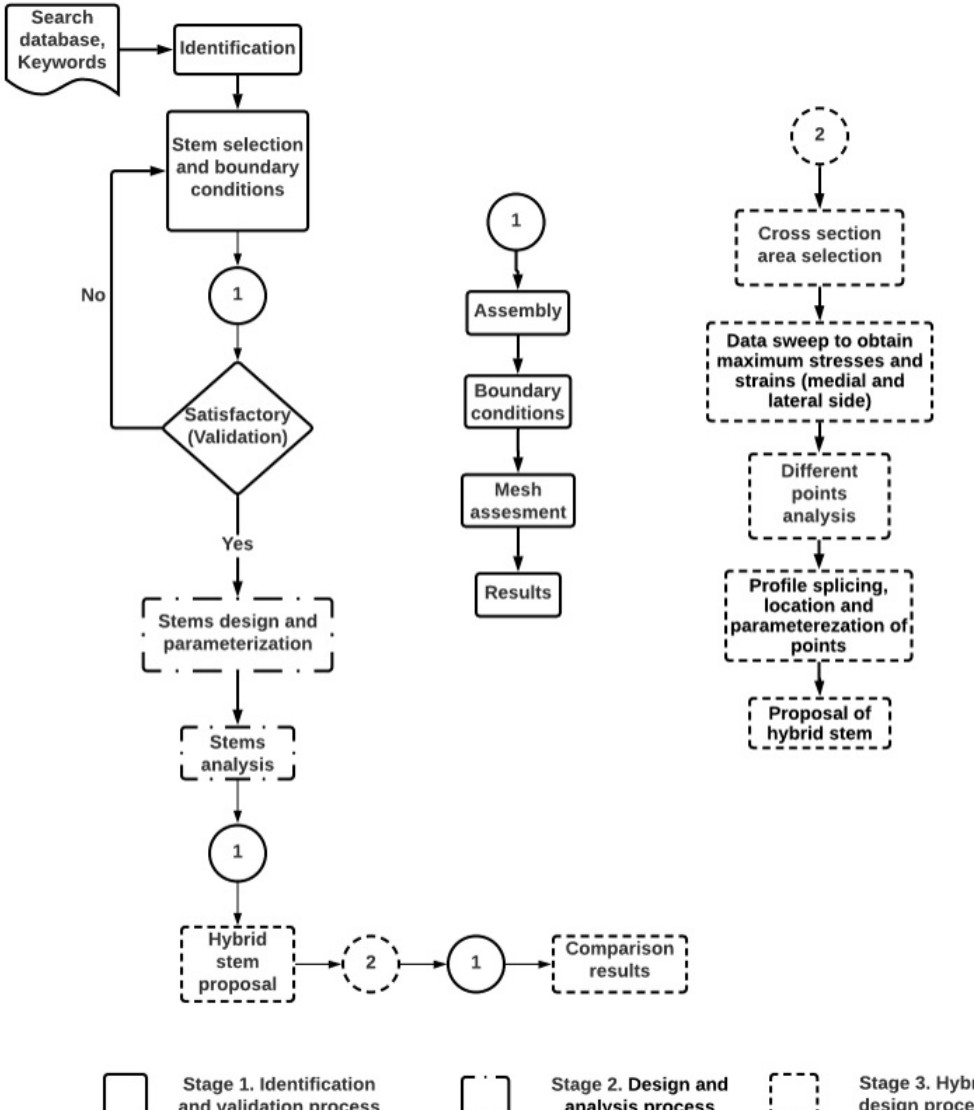

**Figure 1.** Assessment methodology to set a standard base for stem comparison and generate the proposed design of a hybrid prosthesis. It was divided into three stages: Stage 1. Identification and validation process. Stage 2. Design and analysis of the stems. Stage 3. Hybrid design process.

### 2.1. Stage One: Identification and Validation Process

2.1.1. Identification

At this stage, journal papers and databases (ScienceDirect, PubMed, Springer, Scopus) were consulted from the year 2008 to 2022 to identify the most relevant research in areas relating to hip implants, optimization of hip implants, hip implant design, composite hip implant, and stress shielding. Studies regarding different boundary conditions (forces, fix support, types of contacts, and meshing parameters) commonly used in the THR process were identified and are presented in Table 1.

The most common materials, the cross-sectional stem areas (circular, ellipse, oval, and trapezoidal), and the majority of the dimensions (offset, length neck, length stem, cross-sections, radii, angles) to replicate the prostheses (drawing and parametrization) were also identified in this section.

**Table 1.** Results from identified stems: boundary conditions and meshing parameters.

| Author and Year | Stems | Boundary Conditions | | | | Meshing Parameters | | | |
|---|---|---|---|---|---|---|---|---|---|
| | | Force | | Fix Support | Type of Contacts | Type of Element | Number of Nodes | Number of Elements | Element Size |
| Wen-Chen et al., 2014 [42] | Stem size No. 15 Versys | Femoral head (N) $F_x = 1492$ $F_y = 915$ $F_z = -2925$ | Abduction muscle (N) $F_x = -1342$ $F_y = -832$ $F_z = 2055$ | Distal end for the cortex | Stem-bone | 20-node tetrahedron | — | 68,311 | — |
| | Size number 7 ABG | Femoral head (N) $F_x = 1492$ $F_y = 915$ $F_z = -2925$ | Abduction muscle (N) $F_x = 1492$ $F_y = 915$ $F_z = -2925$ | Distal end for the cortex | Stem-bone | 20-node tetrahedron | — | 71,457 | — |
| Faizan et al., 2015 [21] | Accolade TZMF (Stryker) | Femoral head (N) $F_X = 652$ $F_y = 413$ $F_z = -1398$ | | Fixed boundary conditions were applied at a vertical distance | Node to surface augmented lagrangian non-linear contact (0.35) | — | — | — | — |
| | Accolade TZMF (Stryker) | | | | | — | — | — | — |
| Bougherara et al., 2011 [29] | A novel composite | 3 kN | | Fixed distal base | Bonded contact between contact surfaces | 10-node quadratic tetrahedron | 171,531 | 134,769 | — |
| | Exeter (Stryker) | | | | | | 314,658 | 240,272 | — |
| | Omnifit (Stryker) | | | | | | 149,972 | 106,940 | — |
| Restrepo et al., 2011 [43] | Restoration modular stem (Stryker) | — | | — | — | — | — | — | — |
| Nandi et al., 2011 [10] | New design | Vertical load of 3560 N | | Fixed distal | — | — | — | — | — |
| Barahuddin et al., 2014 [31] | New design | Normal walking (Hip contact) $F_X = -378$ $F_y = -229.6$ $F_z = -1604.1$ | Stairs climbing (Hip contact) $F_X = -415.1$ $F_y = -424.2$ $F_X = -1654.1$ | Completely restrained distally | Deformable contact between stem and femoral | Tetrahedral | 5000 | 34,000 | 0.4 mm |

**Table 1.** *Cont.*

| Author and Year | Stems | Boundary Conditions | | | Meshing Parameters | | | |
|---|---|---|---|---|---|---|---|---|
| | | Force | Fix Support | Type of Contacts | Type of Element | Number of Nodes | Number of Elements | Element Size |
| Gkagkalis et al., 2019 [44] | Fitmore and Allofit | — | — | — | — | — | — | — |
| Braileanu et al., 2018 [15] | New design | 4200 N (Femoral head) | Distal end to the femur and stem | — | — | — | — | — |
| Delikanil et al., 2019 [45] | New design | 300 N to 2300 N normal to the neck | The bottom plane of the cement | Bonded and no separation contacts | 10 and 20 node tetrahedral | — | 65,000 (solid implants) | — |
| Rezaei et al., 2015 [46] | Design based on Oshkour et al., 2014 [2] | 3 KN at an angle of 20° | Distal end of the bone | — | Metalic: Wedge-triangular | 16,254 | 29,707 | — |
| | | | | | Composite: cubic | 10,187 | 8613 | |
| | | | | | Bone: Wedge-triangular | 42,014 | 71,033 | |
| Mohamed et al., 2018 [47] | New design | — | Distal end of the stem. | — | — | — | — | 0.004 mm |
| Sabatini and Goswami 2008 [9] | New design | 3000 N normal to the neck of the implant | Fully fixed at the distal end of the femur and partially at the proximal end near the greater trochanter | Bonded | | | 60,000 | |
| Oshkour et al., 2014 [2] | Design based on Sabatini and Goswami 2008 [9] | 3000 N at an angle of 20° | Distal end of the femur | Surface to surface contact property with finite sliding and friction coefficient of 0.3 bone-prosthesis interface | Tetrahedral | — | — | 1.5 mm |

**Table 1.** *Cont.*

| Author and Year | Stems | Boundary Conditions | | | Meshing Parameters | | | |
|---|---|---|---|---|---|---|---|---|
| | | Force | Fix Support | Type of Contacts | Type of Element | Number of Nodes | Number of Elements | Element Size |
| Ro et al., 2018 [14] | New Design | 1.8–3.2 kN | Distal end of the femur | — | — | — | — | — |
| K.N. et al., 2019 [28] | Design based on Oshkour et al., 2014 | 2300 N normal to the neck | Middle to distal end of the stem | — | unstructured mesh | 675,000 | 495,000 | 1 mm |

### 2.1.2. Parametrization of the Representative Stem

A representative stem (Exeter) was selected from the stems in Table 1, which was accomplished with the criteria of full reported results, precise geometric dimensions, validation through experimental work, and smooth geometry [29,48].

The implant profile was replicated and parameterized in Solidworks 2017. The Exeter V40 N° 2 (Stryker Corporation, Mahwah, NJ, USA [48]) is reported to have a total length of 150 mm, 37.5 mm offset, and a neck length of 45.8 mm. Different planes were created to draw the Exeter stem (Figure 2a)) while focusing on four specific places of the stem profile: distal, proximal, neck, and the top section of the implant (where the femoral head is located). Those planes acted as a reference to create the cross-sectional area of the implant. After creating the cross-sectional area, the loft tool from the CAD software was used to generate the prosthesis' 3D shape. Finally, to avoid errors, all the generated surfaces were filled and knitted together to develop the solid part [2,9,28,29,48]. After the stem was parametrized, it was assembled in a concrete block (88 × 88 × 80 mm high, Bougherara et al. [29]) to simulate the prosthesis embedding into the bone (Figure 2b)). The contact between the distal region of the implant and the support block was set, and on the proximal section of the stem, the femoral head fit precisely.

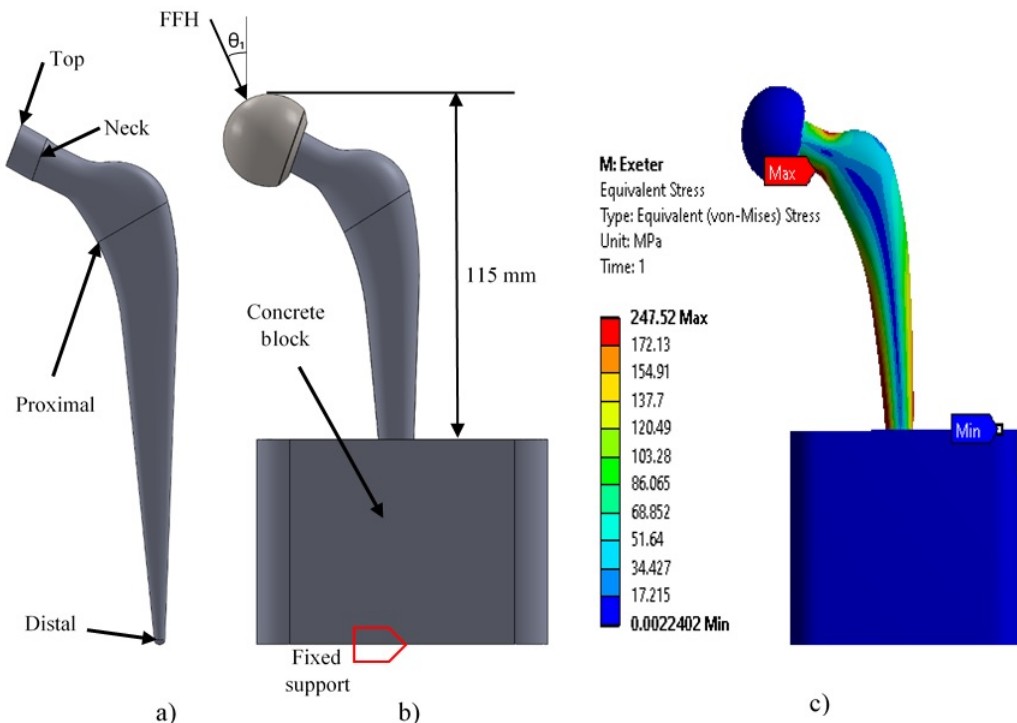

**Figure 2.** (**a**) Lateral view of the Exeter stem. (**b**) Assembly (block-femoral head) and boundary conditions. (**c**) Maximum stress results obtained in Exeter stem.

### 2.1.3. Boundary Conditions Setting and Validation

ANSYS Workbench 2017 was used for the implant analysis. The distal section of the prosthesis and the base of the block were fixed. The contacts between the block–implant and the femoral head–implant were set as: bonded and no separation, respectively (Figure 2b), and the mechanical properties used are described in Table 2.

A 3 kN load simulating the load corresponding to an average person of 75 kg walking was applied at the top of the femoral head for the reproduction analysis, which corresponds to the first activity of Table 3. The 10-node quadratic tetrahedron elements were used with linear formulation for the supporting block, the femoral head, and all the stems assessed. This element has been previously used because of its capacity to perform assessments on

discontinuous solids, aided by the three degrees of freedom in their nodal-translation of each node: x, y, and z directions [2,15,29,31].

**Table 2.** Mechanical properties of the implants and the assembly components [1,11,29,49,50].

| Material | Modulus of Elasticity (GPa) | Poisson Coefficient |
|---|---|---|
| SS316L | 220 | 0.3 |
| CoCrMo | 210 | 0.3 |
| Ti6Al4V | 114 | 0.3 |
| CoCr | 200 | 0.3 |
| Concrete | 30 | 0.18 |

**Table 3.** Loads for different and daily activities. The weight of 75 kg was used for activity 1 ([2,29,46] and 80 kg for activities 2–6 [20,51]). Force on the femoral head (FFH): corresponds to the resultant reaction force (N) acting on the femoral head.

| Number | Activity | Angle | FFH (N) |
|---|---|---|---|
| 1 | Normal walking (75 kg) | $\theta_1 = 20$ | 3000 |
| 2 | 1 leg stand | $\theta_1 = 13$ | 1938.36 |
| 3 | Normal walking (80 kg) | $\theta_1 = 13$ | 5242.26 |
| 4 | Down stairs | $\theta_1 = 12$ | 5015.96 |
| 5 | Knee bend | $\theta_1 = 16$ | 5503.63 |
| 6 | 2-1-2 legs stand | $\theta_1 = 7$ | 2757 |

2.1.4. Meshing Results and Validation.

A mesh analysis was performed over the chosen stem by varying the element size in a 0.1 mm ratio, as previously reported [2,28], until the variation in the obtained results was minimum and stable; see Figure 3. Then, the mesh size was selected, and its results were compared to those published by [29].

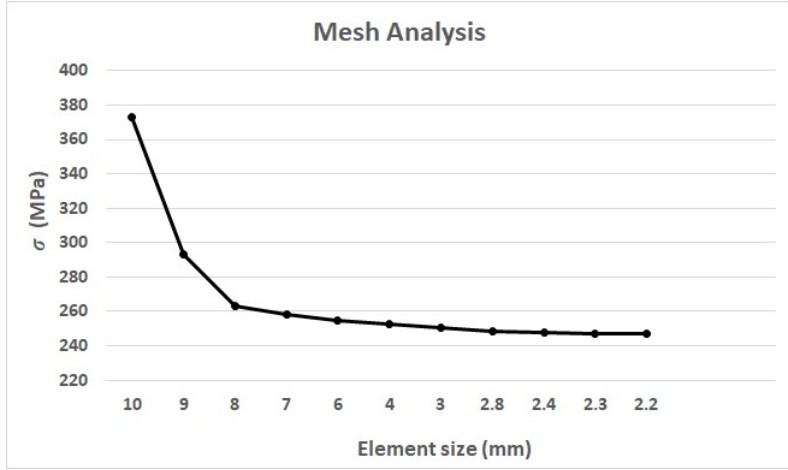

**Figure 3.** Mesh analysis results between the element size and the maximum equivalent stress ($\sigma$) of the Exeter stem.

Bougherara et al. [29] reported 240,272 elements and 314,658 nodes in their work, resulting in the maximum equivalent stress of von Mises of 247.8 MPa. An element size of 2.4 mm (Figure 3) gives (in our analysis) a von Mises stress of 247.52 MPa, generating 181,320 elements and 259,863 nodes, representing an 0.11% error when compared with Bouguerara et al. [29]. The maximum stress for both assessments was located in the same area (medial-neck area), which validates the results obtained here. Therefore, for this work, a 2.4 mm element size, 10-node quadratic tetrahedron elements, the material and loads from

Tables 2 and 3, and the assembly features and the types of contacts described in Section 2.1.3, were used for the prosthesis simulation and assessment of the remaining stems.

### 2.2. Stage Two: Design and Analysis Process

2.2.1. The Stem Design and Parameterization

Sixteen prosthesis configurations were drawn, parameterized, and assessed following the same process described above. Oshkour et al. [2], and Sabatini et al. [9] presented different prostheses that handle the most common transverse areas (circular, oval, ellipse, and trapezoidal) used in the design of the implants. When combined with three different profiles, twelve different configurations were obtained (Table 4). The Accolade II (size II) [21], the modular restoration stem (Stryker) [43], and the Braileanu et al. [15] designs were also reproduced, and their results are presented in Appendix A. The Exeter [29,48] prosthesis was also used in the comparison. The equivalent stress and deformation ranges generated by all stem geometries (distal, medial, and proximal parts) were analyzed.

**Table 4.** Different prosthesis designs and cross-sectional areas based in the information presented by Sabatini et al. [9] and Oshkour et al. [2]. P1 means Profile 1, P2 means Profile 2, and so on, and their corresponding cross-sectional areas are indicated (circular, ellipse, oval, and trapezoidal). In the last row, the different views of the cross-section areas in the proximal part of the stem from Profile 1 can be seen.

| Profile | Circular | Ellipse | Oval | Trapezoidal |
|---------|----------|---------|------|-------------|
| P1 |  |  |  |  |
| P2 |  |  |  |  |
| P3 |  |  |  |  |
| P1 cross sectional view |  |  |  |  |

2.2.2. Stems Analysis and Results

The FEM assessment of all stems and their respective assemblies was carried out using all the boundary conditions and mesh described on Sections 2.1.3 and 2.1.4, respectively. The different loads shown in Table 3 were also used to assess the behavior of the stems in various and typical daily activities (activity 1 was based on a 75 kg person weight, whereas activities 2–6 were based on an 80 kg person weight). This helped us to determine the stem's behavior in different activities instead of restricting the assessment to only one activity [2,9,10,15,21,28,29,42,45]. To maintain similarity between the amount of volume of the stem fitted into the block for all the assemblies, the distance between the highest part of the femoral head assembled with the implant and the upper part of the concrete block was set to 115 mm (see Figure 2b), as reported in [29]. The complete results of the simulations can be seen in Appendix A.

*2.3. Stage Three: The Hybrid Stem Design Process*

2.3.1. Hybrid Stem Proposal

The stress and strain behavior through the stem lengths were studied to generate the hybrid stem proposal; the results are presented in Appendix A. The results indicate that the trapezoidal shape was the most suitable cross-sectional area; it also presented the lower deformation and stress in four of six activities.

The stress and strain behavior of sixteen proposed points created along the stem length were assessed to obtain an optimized geometry based on the best results of the generated points, using the most critical activity as a reference (Down stairs). The stress and strain results of the reference points are presented in Appendix B. A summary of the stems with lower stress results obtained from Appendix B is shown in Table 5. Those results were used to parametrize the hybrid proposal (shape and cross-sectional area) by overlapping their configurations.

**Table 5.** Summary of the best results (lower values in stress and strain) of the generated reference points on the lateral and medial side of all stems from Appendix B. P1, P2, and P3 represent the profile numbers, and the letter represents the type of cross-section (T = trapezoidal, C = circular). The letters "M" and "L" stand for the medial and the lateral parts of the stem. The same boundary conditions and the mesh used in Sections 2.1.3 and 2.1.4 were used for the FEM analysis.

| | | | | | | | | | | | | | | | |
|---|---|---|---|---|---|---|---|---|---|---|---|---|---|---|---|
| **Maximum Equivalent Stress (von Mises)** | | | | | | | | **Maximum Equivalent Strain (von Mises)** | | | | | | | |
| **Medial** | | | | **Lateral** | | | | **Medial** | | | | **Lateral** | | | |
| **Stem** | **Location** | **σ (MPa)** | **Material** | **Stem** | **Location** | **σ (MPa)** | **Material** | **Stem** | **Location** | **ε (mm/m)** | **Material** | **Stem** | **Location** | **ε (mm/m)** | **Material** |
| P2-T | M1 | 340.08 | SS316L | P2-T | L1 | 189.18 | SS316L | P2-T | M1 | 1.55 | SS316L | P2-T | L1 | 0.86 | SS316L |
| P2-T | M2 | 238.28 | SS316L | P2-T | L2 | 125.65 | SS316L | P2-T | M2 | 1.08 | SS316L | P2-T | L2 | 0.57 | SS316L |
| P3-T | M3 | 153.89 | SS316L | P1-T | L3 | 41.82 | SS316L | P3-T | M3 | 0.70 | SS316L | P1-T | L3 | 0.19 | SS316L |
| P3-T | M4 | 114.30 | SS316L | P3-T | L4 | 13.74 | SS316L | P3-T | M4 | 0.52 | SS316L | P3-T | L4 | 0.06 | SS316L |
| P1-T | M5 | 90.34 | SS316L | P2-T | L5 | 12.22 | SS316L | P1-T | M5 | 0.41 | SS316L | P2-T | L5 | 0.06 | SS316L |
| P1-T | M6 | 145.28 | SS316L | P2-T | L6 | 86.33 | SS316L | P1-T | M6 | 0.66 | SS316L | P2-T | L6 | 0.39 | SS316L |
| P1-T | M7 | 205.28 | SS316L | P1-T | L7 | 141.37 | SS316L | P1-T | M7 | 0.93 | SS316L | P1-T | L7 | 0.64 | SS316L |
| P1-T | M8 | 268.88 | SS316L | P1-T | L8 | 206.98 | SS316L | P1-T | M8 | 1.22 | SS316L | P1-T | L8 | 0.94 | SS316L |

2.3.2. Generating a Stem Proposal

The multi-objective analysis applied to generate the stem proposal was carried out using 16 reference points (eight on the medial side "M", and eight on the lateral side "L") placed on both sides of the hip implant. To set the reference, four planes containing the lines M1-L1, M3-L3, M5-L5, and M7-L7, were created around an angle of 35° using the baseline, and parallel lines from it, replicating the views presented by Sabatini et al. [9]; the remaining planes (M2-L2, M4-L4, M6-L6, and M8-L8) were placed in the middle of the

planes mentioned above. The numbering of the points starts at the reference line located at the block's top surface and goes up to the stem neck (Figure 4a). As a result, new points were generated (Figure 4b) on both the medial and lateral sides. The next step was to place the new reference points in a plane using their coordinates, corresponding profiles and cross-sectional areas (Table 5). Finally, the hybrid profile (Figure 4c) was drawn by linking the new reference points described above. Those points aided in defining the radii and angles of the profile to generate the solid. It is important to highlight that the trapezoidal cross-sectional area was the geometry with the best results, which agrees with the findings of Appendix A and the results reported by Oshkour et al. [2], Sabatini et al. [9], and C. KN et al. [28].

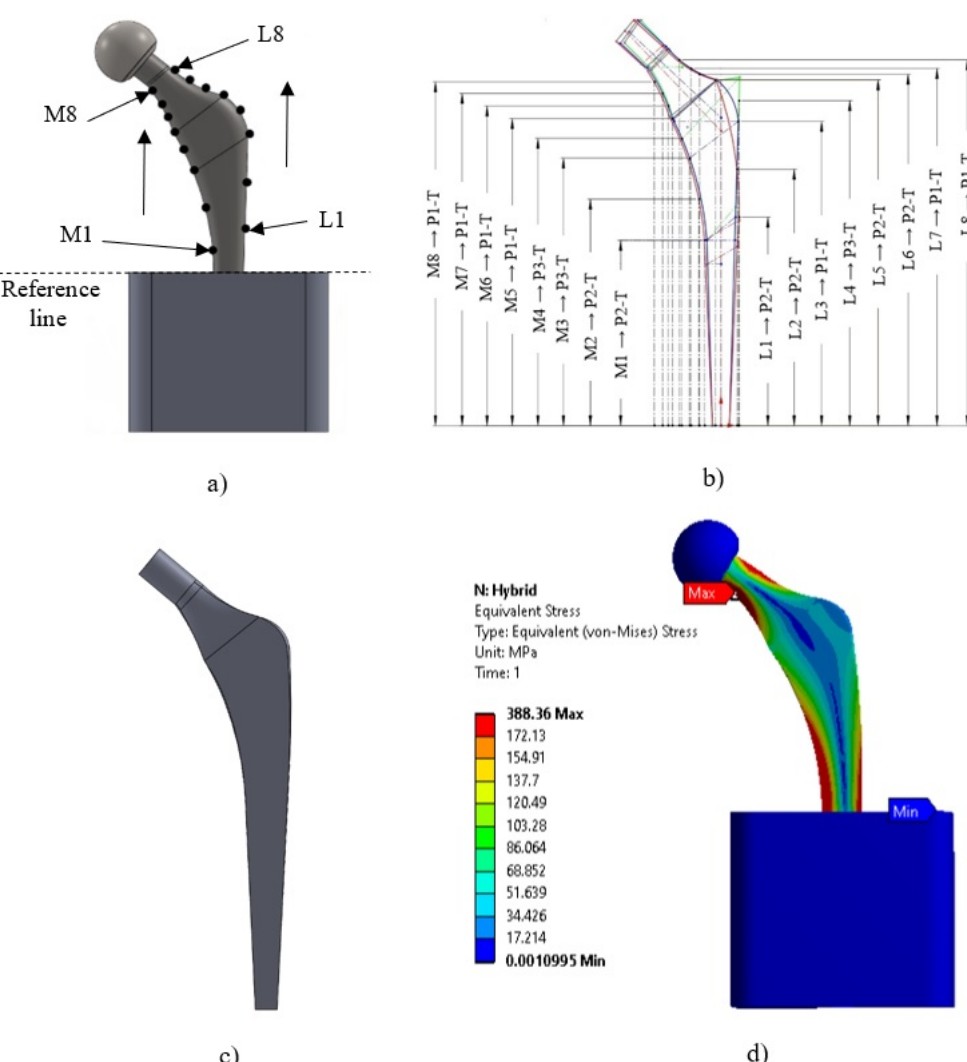

**Figure 4.** (**a**) Locations of 8 points in the medial side and 8 points in the lateral side of the implant. (The number of points increases upwards in both parts of the implant, M1 "Medial 1" and L1 "Lateral 1"). (**b**) Locations of the points that showed the best stress and strain results of the different stems evaluated, where (MX/LX → PX-T) represents the path assessed at each of the generated reference points (M1-M8 and L1-L8); P1, P2, and P3 represent the profile numbers; and the letter represents the type of cross-section (T = trapezoidal). (**c**) Design implant proposal. (**d**) Maximum stress with the hybrid profile using the load of activity 4 (Down stairs) and the stainless steel material (SS316L).

### 2.3.3. Analysis with ASTM F2996-13 and ISO 7206–4:2010(E)

Finally, another study was carried out on the hybrid implant using the ASTM F2996-13 (see Figure 5) and the loading conditions of the ISO 7206–4: 2010 (E) [28], to observe the performance of this design with respect to the other configurations (P1, P2 and P3 with the circular cross-sectional area, ellipse, oval, and trapezoidal).

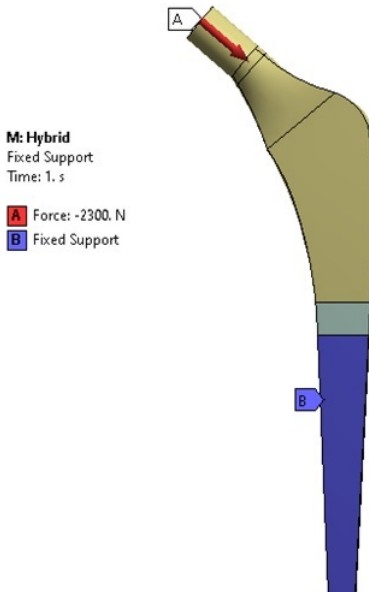

**Figure 5.** Boundary conditions considered as per ASTM F2996-13 standards.

## 3. Results

Comparing the stems boundary conditions and mesh sizing published in the literature is complicated because of the wide variety of studies that focus on different subjects and the limited access to the drawings of stem manufacturers (see Table 1). From this Table, the Exeter prosthesis [29,48] was selected because it presented complete information about dimensions.

The first step was to determine and validate the boundary conditions indicated in Section 2.1 and used to reproduce the work of Bougherara et al. [29]. As a result, 0.11% error was obtained between the stress reported by Bougherara et al. [29] (247.8 MPa) and the results obtained from this work (247.52 MPa). The stresses location were consistent in the same area for both studies, showing the reliability of the results obtained from this work.

The multi-objective analysis (equivalent stress and strain of von Mises) in the sixteen implants was assessed under the same conditions, and their results are shown in Appendix A. According to Appendix A, the maximum von Mises stress and strain were located in the medial part of the stem between the M1 point and the reference line (top surface of the concrete block, Figure 4a). Activity 1 shows the maximum stress and deformation in the medial neck section of the hip implant around M8 in Figure 4a).

The lower results obtained from the 16 points created in Section 2.3.2 were used to generate the new geometry for the hybrid profile. The trapezoidal cross-section geometry presented the most consistent behavior for both stresses and strains (Appendix B) and was used as a base for the proposal. The sixteen chosen reference points of Table 5 were overlapped and re-parametrized to obtain a hybrid stem geometry (Figure 4c).

The hybrid proposal underwent the same analysis process described previously, and it is compared with the best stem (reference stem, P2-T) of Appendix A in Table 6. The hybrid profile produced lower stress results when the applied load had an angle between 7°–13° (activities 2–4 and 6). While for activities 1 and 5 (20° and 16°), P2-C produced the lower stress results. The hybrid profile presents the best results for all the activities in terms of deformation. The maximum stress on the hybrid profile occurred in the medial neck area

for activities 1 to 5 (Figure 4d)) and, in the medial base area for activity 6. Regarding the maximum strain, it was found in the medial base zone for almost all the activities, except on activity 1 where the maximum strain was located in the medial neck zone.

The results of the different stems assessed with ASTM F2996-13 and ISO 7206–4:2010 (E) [28] are presented in Figures 6 and 7; note that the hybrid design shows the lowest equivalent von Mises stress and strain.

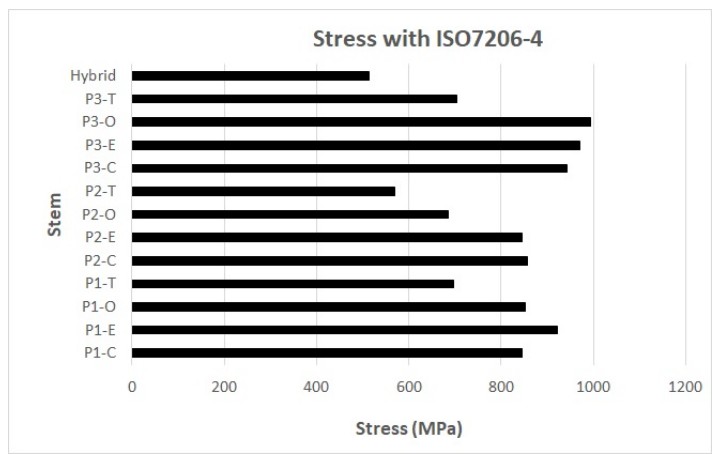

**Figure 6.** Results of equivalent stress of von Mises with the use of ASTM F2996-13 and ISO 7206–4:2010(E). P1, P2, and P3 represent the profile numbers, and the letter represents the type of cross-section (C = circular, E = ellipse, O = oval, T = trapezoidal).

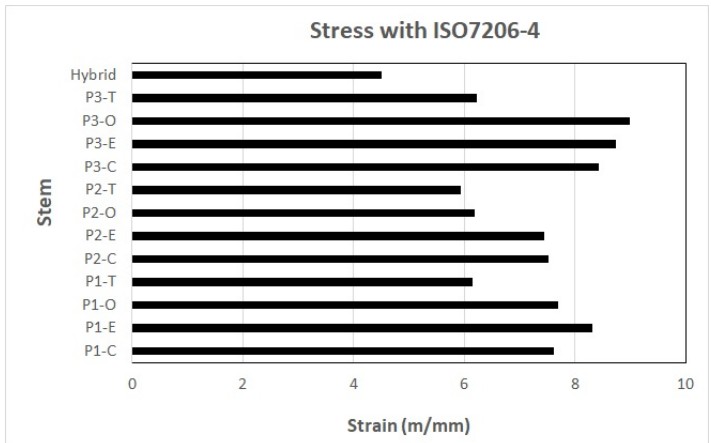

**Figure 7.** Results of equivalent strain of von Mises with the use of ASTM F2996-13 and ISO 7206–4:2010(E). P1, P2, and P3 represent the profile numbers, and the letter represents the type of cross-section (C = circular, E = ellipse, O = oval, T = trapezoidal).

**Table 6.** Comparison of results between the best configurations analyzing the lower equivalent stresses and strains. Here, P2 represents the profile number. The locations (L) of the stresses and strains are given by MN = medial neck and MB = medial base.

| Analyzed Stem | Material | Activity 1: Normal Walking (70 kg) | | | | Activity 2: 1 Leg Stand | | | | Activity 3: Normal Walking (80 kg) | | | | Activity 4: Down Stairs | | | | Activity 5: Knee Bend | | | | Activity 6: 2-1-2 Leg Stand | | | |
|---|---|---|---|---|---|---|---|---|---|---|---|---|---|---|---|---|---|---|---|---|---|---|---|---|---|
| | | σ (MPa) | L | ε (mm/m) | L | σ (MPa) | L | ε (mm/m) | L | σ (MPa) | L | ε (mm/m) | L | σ (MPa) | L | ε (mm/m) | L | σ (MPa) | L | ε (mm/m) | L | σ (MPa) | L | ε (mm/m) | L |
| P2-Trapezoidal | SS316L | 188.28 | MN | 1.50 | MB | 153.51 | MB | 2.18 | MB | 405.94 | MB | 5.75 | MB | 415.69 | MB | 5.93 | MB | 378.85 | MN | 4.63 | MB | 302.31 | MB | 4.41 | MB |
| | CoCrMo | 188.27 | MN | 1.53 | MB | 154.20 | MB | 2.23 | MB | 407.77 | MB | 5.88 | MB | 417.56 | MB | 6.06 | MB | 378.83 | MN | 4.74 | MB | 303.67 | MB | 4.51 | MB |
| | Ti6Al4V | 188.18 | MN | 2.10 | MB | 165.27 | MB | 2.97 | MB | 437.05 | MB | 7.85 | MB | 447.53 | MB | 8.08 | MB | 378.65 | MN | 6.37 | MB | 325.43 | MB | 5.97 | MB |
| Hybrid | SS316L | 194.58 | MN | 0.89 | MN | 150.53 | MN | 1.12 | MB | 398.06 | MN | 2.97 | MB | 388.36 | MN | 3.08 | MB | 392.50 | MN | 2.36 | MB | 285.49 | MB | 2.32 | MB |
| | CoCrMo | 194.58 | MN | 0.93 | MN | 150.52 | MN | 1.15 | MB | 398.05 | MN | 3.03 | MB | 388.36 | MN | 3.14 | MB | 392.49 | MN | 2.41 | MB | 286.69 | MB | 2.36 | MB |
| | Ti6Al4V | 194.53 | MN | 1.77 | MN | 150.49 | MN | 1.49 | MB | 397.97 | MN | 3.94 | MB | 404.43 | MB | 4.07 | MB | 392.41 | MN | 3.58 | MN | 302.05 | MB | 3.07 | MB |

## 4. Discussion

Several studies have assessed the behavior of different hip prostheses under loads and boundary conditions (Table 1). Comparing them represents a challenging task. Even though most studies agree that the neck area is where the stress and strain values concentrate [2,9,14,45,52], those results are functions of the activity assessed, which vary from study to study, as well as their boundary conditions.

This research evaluated the most common hip stem geometries reported in the literature (Table 1) using a multi-objective analysis under the same criteria, enabling the comparison between them. The selected Exeter stem showed most of the geometry dimensions required for its parametrization [29,48], including complete experimental validation reports using strain gauges. It has a smooth geometry without sharp edges, which minimizes the stress concentration due to abrupt geometry changes.

The 10-node quadratic tetrahedron elements are commonly used for the FEM assessment on irregular and curved geometries [2,15,29,31]. This work agrees with those and considered the 10-node quadratic as the best option for this type of study. The authors suggest the use of a uniform size element for similar geometries will aid in comparing the results; the 2.4 mm size used in this study worked very well for all the implant assessments. However, a smaller element size would be required for smaller stem geometries (pediatric stems) or complex stem geometry. Even though some authors report smaller element sizes ranging from 0.004 to 1.5 mm [2,28,31,47], in this work, the selected element size (2.4 mm) gives the lowest percentage of error after comparing it with the outcome of Bougherara et al. [29], by comparing the stems only. Using smaller values could represent a disadvantage in data processing, mainly if the bone–stem interface is considered in the simulation or if cartilage analysis is desired [53–56].

This work focused only on the stem behavior and not on the interaction of bone–stem, simplifying the assessment by using regular geometries as the block used in the assembly. For similar studies, simple geometries have been used [29,45,52]. A concrete block with an elasticity modulus of 30 GPa like the one used here has a very similar elasticity modulus to the human bone previously reported [8,11,12,16], which gives us a reasonable idea of the bone's reaction. The femoral head is commonly used for the load application [14,21,28,29,42,45,47], and it is considered the best zone by the authors to place the load because it resembles the mechanisms of action of the prosthesis at the hip joint.

The trapezoidal cross-section area showed the most significant decrease in stress within the range of 1.96–9.6% in four of the six activities, which agrees with the results given by Oshkour et al. [2], Sabatini et al. [9], and C.K.N. et al. [28]. They reported that the trapezoidal area showed the best performance using the multi-objective analysis. In addition, it showed the most consistent cross-sectional area behavior along the stem (Table 5). Using a trapezoidal cross-sectional area improves the stability of the bone–implant junction because a geometry with pronounced edges restricts the motion and acts as an anchor, which increases the possibility of avoiding micro-movement.

Analysis of the different cross-sections exhibited that the trapezoidal cross-section had a higher effect on reducing micro-movements than smother geometries, such as circular, elliptical, and oval ones [9,57]. This finding agreed with the study by Delikanli and Kayacan [45], where it was found that the femur stability increases with the use of rectangular and trapezoidal sections. Profile 2 with the trapezoidal cross-section (P2-T) showed the lowest stress results in four out of six activities (Appendix A), which are the most likely activities to be repeated every day [28], including "Down stairs", which was the most critical activity identified for the stems (activity 4). When P2-T results are compared with the results of other cross-sectional areas in activity 4 from Appendix A, it can be appreciated that:

The highest stresses were obtained on the elliptical cross-sectional area profiles. A stress increment of 96.6% and a strain decrement of 4.9% between P2 trapezoidal and the elliptical shapes (P1-E) can be appreciated. Those results match with those of the study

presented by Sabatini et al. [9], where the circular and elliptical designs did not show lower stress values compared with the trapezoidal ones.

Results of the circular cross-sectional area showed that it was more sensitive to the force angle application (activities 1 and 5, force angles of 20 and 16 degrees, respectively). This can be the consequence of the increase in the horizontal component of the force vector. In addition, when the implant has a smoother cross-section, the stress is better distributed around the implant. There will also be a reduction in the stress concentration, as mentioned by Sabatini et al. [9] and Kayabasi et al. [57].

Regarding the strains, the distal part and the neck of the implant are the zones with the smallest cross-section area, the last one being more critical in profiles with a circular cross-sectional area. Finally, the oval cross-sectional area profiles showed a maximum stress increment of 75.4% (P3-O) and a strain decrement of 19.7% (P1-O) with respect to P2-trapezoidal.

With softer profiles (with a longer radius on the lateral part of the implant) and rigid materials, a better distribution of the stresses was obtained, generating less deformation. However, the risk of micro-movements could increase [2].

*Hybrid Proposal*

The hybrid profile in activities 2–4 and 6 showed stress reductions of 1.96–9.6% (see Table 6) for the different activities when compared with the reference stem (P2- trapezoidal); this reduction was possible by combining the characteristics of the stems with better results obtained from the multi-objective analysis. The maximum stress in the hybrid profile occurred in the medial neck area for the first five activities.

The lowest strain in all the activities was obtained from the hybrid profile. The mechanical behavior of the hybrid stem suggests a decrease in the deformations at the prosthesis base for all the evaluated loads. A reduction in strains is a potential advantage for patients with osteoporosis because fractures cannot occur with small strains [58,59].

An increment of 5.22 mm in the proximal lateral dimension of the cross-section area (dimension between point M5 and L5 of the same plane) of the hybrid profile was obtained due to the new geometry (the final dimensions were 27.57 mm). Despite modifications in the dimensions around the middle and top area of the stem, the distal section embedded in the bone did not suffer any change. The increment in the dimensions around the proximal section could be a potential limitation for the end-user because it will depend on the particular morphology of the potential user; however, it does not seem to affect the implantation of the stem in the bone. According to the studies presented by Eckrich et al. [60], the femur has dimensions of 33.8 ± 1.3 mm in the proximal part (medullary transverse diameter in the lesser trochanter). The work presented by Pi et al. [61] said that the femur had dimensions of 25.47 ± 3.60 mm, which guarantees that the hybrid implant must fit in the femur. However, it could be limited to the specific anatomical restrictions of each patient. This increment (5.22 mm) will be useful in overweight patients, according to Takai et al. [62], which suggests that a thin stem (neck and body) is not the best option for these patients.

Improvement in the stability and the fixation of the stem in the bone are the main concerns for the surgeons during THR [42]. The hybrid design proposes an increment in the proximal area and the advantage of being an almost straight implant with a trapezoidal cross-sectional area that will aid in the stem stability and fixation.

A stem elasticity modulus that approximates as much as possible the modulus of elasticity of the bone is desirable. Less deformation is obtained with a rigid material; therefore, the load distribution is smaller towards the spongy bone. Oshkour et al. [2] described that in the neck area of the implant, the stress behavior is practically the same; either stainless steel (SS316L) or titanium (Ti6Al4V) can be used. This behavior is caused because it is the area where the implant does not contact the bone. Therefore, it does not distribute the load to the bone. This behavior could be observed in the different prosthesis configurations assessed in this research. Currently different metals are used for the femoral componente of the THR and their muduli of elasticity is higher than cortical bone, therfore

potential problems could be generated, such as aseptic loosening [29]. SS316L is the material with the best strain results, and titanium is a better option in terms of stress but is more expensive.

Currently, ASTM F2996-13 and ISO 7206–4:2010(E) are the norms used for the hip implant assessment. However, differences in the loads and boundary conditions between the norms and the methodology used in this work seem to make the comparison difficult. To perform the comparison, the percentage difference in stress between the hybrid and the reference stem was used (Table 6). As a result, the hybrid implant showed a stress reduction of 9.6% compared to the reference implant (P2-T) in activity 4, considering the titanium alloy (Ti6Al4V) as the material and the methodology used for the authors of this work. Additionally, with ASTM F2996-13 and ISO 7206–4:2010(E) standards, a stress reduction of 9.98% was obtained (Figure 6). This shows concordance with the results obtained, even with the different methodologies.

On the other hand, concerning the strain, a decrease of 24.1% was obtained with the use of ASTM F2996-13 and ISO 7206–4:2010(E) when compared to the reference implant (P2-T) in activity 4 using the titanium alloy (Ti6Al4V, see Figure 7).

Future work to upgrade the performance of the hybrid implant will be the optimization of the cross-sectional area, and the use of new materials, such as composite materials, to improve the distribution of the stresses and strain in the interaction between the implant and the cortical bone.

## 5. Conclusions

The design requirements of a hip implant are challenging due to many variables that have to be considered, such as the biocompatibility of the materials, the geometry, the bone anatomy, and the patient age; these variables play an important role in the generation of stress and strain in the implant. Designing a new prosthesis by doing a geometry modification or using an optimal biomaterial can be a solution to reduce potential problems, such as aseptic loosening or a fracture of the implant.

This work evaluated sixteen different hip implant configurations found in the literature under multi-objective analysis using a common standardized condition and considering the potential and daily activities performed by the patients. In conclusion, by comparing all the stems using the same criteria, the advantages and disadvantages between the stems can be evaluated. It was found that the use of softer profiles (with a longer radius on the lateral part of the implant) and rigid materials provided better distribution of the stresses and generated less deformation. However, the risk of micro-movements could increase due to the lack of edges that increase the contact area. If this problem needs to be reduced, a trapezoidal cross-sectional area is recommended.

The new geometry was proposed using a multi-objective analysis, based on a trapezoidal cross-section, with a stress reduction in the proximal area and a strain decrease in the whole stem. The hybrid proposal presented a stress reduction of 9.6% compared with the reference stem (P2-T) in activity 4 using the titanium alloy (Ti6Al4V) as material. In addition, using ASTM F2996-13 and ISO 7206–4:2010(E) standards, a stress reduction of 9.98% was obtained using the same material. This reduction was achieved by following the assessment conditions described in the work of K.N. et al. [28]. The results show both methods comply with the design requirements.

Stress/strain reduction is a potential advantage for patients with osteoporosis. In terms of materials, SS316L showed the lowest strain results. However, the modulus of elasticity was still very high compared to cortical bone, which could cause various problems such as wear and micro-movement. In contrast, titanium is a better option in terms of stress. Although it is a more expensive option and its modulus of elasticity is lower than SS316L, it is still high compared to cortical bone.

This leads researchers to continue searching for a material that performs an optimal function in the interaction of bone–implant and generates a better design depending

on the morphological needs and health problems that the patient may present (such as osteoporosis or fractures) when performing the surgery.

**Author Contributions:** Formal analysis, M.S. and L.G.; Methodology, I.S.; Project administration, E.D.; Validation, D.J.; Writing—review & editing, M.G. and A.O. All authors have read and agreed to the published version of the manuscript.

**Funding:** This research received no external funding.

**Institutional Review Board Statement:** Not applicable.

**Informed Consent Statement:** Not applicable.

**Data Availability Statement:** Not applicable.

**Acknowledgments:** E.S. Durazo-Romero and A. S. Ortiz-Pérez thank the L. of L., K. of K., and G. of G. The technical support of Jorge Anguiano, Daniel Barrera, Jorge Miramón, Miriam Siqueiros, Luis González Israel Sauceda, and David Jiménez is also acknowledged. M. Guzmán Herrera thanks the scholarship 327470 from CONACyT.

**Conflicts of Interest:** The authors declare no conflict of interest.

## Appendix A

**Table A1.** Stems maximum stress ($\sigma$) and equivalent strain ($\epsilon$), for six different activities and its respective location indicated by MN = Medial Neck, MB = Medial Base.

| Analyzed Stem | Material | Activity 1: Normal Walking (70 kg) | | | | Activity 2: 1 Leg Stand | | | | Activity 3: Normal Walking (80 kg) | | | | Activity 4: Down Stairs | | | | Activity 5: Knee Bend | | | | Activity 6: 2-1-2 Leg Stand | | | |
|---|---|---|---|---|---|---|---|---|---|---|---|---|---|---|---|---|---|---|---|---|---|---|---|---|---|
| | | $\sigma$ (MPa) | L | $\epsilon$ (mm/m) | L | $\sigma$ (MPa) | L | $\epsilon$ (mm/m) | L | $\sigma$ (MPa) | L | $\epsilon$ (mm/m) | L | $\sigma$ (MPa) | L | $\epsilon$ (mm/m) | L | $\sigma$ (MPa) | L | $\epsilon$ (mm/m) | L | $\sigma$ (MPa) | L | $\epsilon$ (mm/m) | L |
| P1-Circular | CoCrMo | 191.41 | MN | 1.66 | MB | 233.04 | MB | 2.73 | MB | 616.25 | MB | 7.21 | MB | 635.19 | MB | 7.48 | MB | 495.96 | MB | 5.64 | MB | 472.54 | MB | 5.69 | MB |
| P1-Ellipse | CoCrMo | 196.37 | MB | 1.28 | MB | 300.29 | MB | 2.11 | MB | 794.11 | MB | 5.59 | MB | 820.75 | MB | 5.80 | MB | 631.78 | MB | 4.37 | MB | 616.31 | MB | 4.42 | MB |
| P1-Oval | CoCrMo | 187.21 | MB | 1.58 | MB | 234.68 | MB | 2.65 | MB | 620.59 | MB | 7.00 | MB | 633.68 | MB | 7.27 | MB | 519.10 | MB | 5.44 | MB | 461.92 | MB | 5.55 | MB |
| P1-Trapezoidal | CoCrMo | 176.62 | MN | 1.22 | MB | 193.05 | MB | 1.95 | MB | 510.51 | MB | 5.15 | MB | 525.98 | MB | 5.34 | MB | 411.74 | MB | 4.05 | MB | 390.75 | MB | 4.06 | MB |
| P2-Circular | CoCrMo | 156.11 | MN | 1.00 | MB | 163.49 | MB | 1.57 | MB | 432.34 | MB | 4.14 | MB | 444.65 | MB | 4.29 | MB | 351.15 | MB | 3.27 | MB | 328.31 | MB | 3.17 | MB |
| P2-Ellipse | CoCrMo | 195.60 | MN | 1.11 | MB | 202.21 | MB | 1.66 | MB | 534.74 | MB | 4.39 | MB | 549.17 | MB | 4.53 | MB | 436.94 | MB | 3.51 | MB | 403.46 | MB | 3.39 | MB |
| P2-Oval | CoCrMo | 195.68 | MN | 1.48 | MB | 191.01 | MB | 2.26 | MB | 505.12 | MB | 5.96 | MB | 518.00 | MB | 6.16 | MB | 416.11 | MB | 4.75 | MB | 379.23 | MB | 4.62 | MB |
| P2-Trapezoidal | CoCrMo | 188.27 | MN | 1.53 | MB | 154.20 | MB | 2.23 | MB | 407.77 | MB | 5.88 | MB | 417.56 | MB | 6.06 | MB | 378.83 | MN | 4.74 | MB | 303.67 | MB | 4.51 | MB |
| P3-Circular | CoCrMo | 186.47 | MN | 1.39 | MB | 219.80 | MB | 2.44 | MB | 581.25 | MB | 6.45 | MB | 600.99 | MB | 6.72 | MB | 462.32 | MB | 4.97 | MB | 451.86 | MB | 5.17 | MB |
| P3-Ellipse | CoCrMo | 225.48 | MN | 1.17 | MB | 246.82 | MB | 1.96 | MB | 652.71 | MB | 5.18 | MB | 675.52 | MB | 5.38 | MB | 528.91 | MB | 4.03 | MB | 509.58 | MB | 4.11 | MB |
| P3-Oval | CoCrMo | 219.39 | MN | 1.18 | MB | 267.97 | MB | 1.99 | MB | 708.64 | MB | 5.27 | MB | 732.31 | MB | 5.47 | MB | 564.14 | MB | 4.09 | MB | 549.63 | MB | 4.19 | MB |
| P3-Trapezoidal | CoCrMo | 214.51 | MN | 1.26 | MB | 197.44 | MB | 2.05 | MB | 522.12 | MB | 5.43 | MB | 539.47 | MB | 5.63 | MB | 431.28 | MN | 4.25 | MB | 404.69 | MB | 4.28 | MB |
| P. Exeter | CoCrMo | 247.52 | MN | 1.98 | MB | 379.00 | MB | 3.47 | MB | 1002.25 | MB | 9.17 | MB | 1035.89 | MB | 9.55 | MB | 797.27 | MB | 7.06 | MB | 777.87 | MB | 7.35 | MB |
| P. Accolade II | CoCrMo | 1183.32 | MB | 26.51 | MB | 641.13 | MB | 14.09 | MB | 1695.46 | MB | 37.25 | MB | 1569.62 | MB | 34.37 | MB | 1950.58 | MB | 43.24 | MB | 715.42 | MB | 15.38 | MB |
| P. Restoration S. | CoCrMo | 842.29 | MB | 9.80 | MB | 483.57 | MB | 5.11 | MB | 1278.78 | MB | 13.51 | MB | 1195.82 | MB | 12.43 | MB | 1431.74 | MB | 15.83 | MB | 578.64 | MB | 5.68 | MB |
| P. Braileanu | CoCrMo | 314.02 | MN | 1.50 | MN | 248.39 | MN | 1.26 | MB | 656.82 | MN | 3.32 | MB | 642.51 | MN | 3.44 | MB | 642.10 | MN | 3.06 | MN | 390.10 | MN | 2.60 | MB |

## Appendix B

**Table A2.** Stress and strain analysis on different points in the medial and lateral side of the stem. (Activity 4: Down stairs).

| Location | Material | P1-Circular | | P1-Ellipse | | P1-Oval | | P1-Trapezoidal | | P2-Circular | | P2-Ellipse | | P2-Oval | | P2-Trapezoidal | | P3-Circular | | P3-Ellipse | | P3-Oval | | P3-Trapezoidal | |
|---|---|---|---|---|---|---|---|---|---|---|---|---|---|---|---|---|---|---|---|---|---|---|---|---|---|
| | | $\sigma$ (MPa) | $\epsilon$ (mm/m) | $\sigma$ (MPa) | $\epsilon$ (mm/m) | $\sigma$ (MPa) | $\epsilon$ (mm/m) | $\sigma$ (MPa) | $\epsilon$ (mm/m) | $\sigma$ (MPa) | $\epsilon$ (mm/m) | $\sigma$ (MPa) | $\epsilon$ (mm/m) | $\sigma$ (MPa) | $\epsilon$ (mm/m) | $\sigma$ (MPa) | $\epsilon$ (mm/m) | $\sigma$ (MPa) | $\epsilon$ (mm/m) | $\sigma$ (MPa) | $\epsilon$ (mm/m) | $\sigma$ (MPa) | $\epsilon$ (mm/m) | $\sigma$ (MPa) | $\epsilon$ (mm/m) |
| M1 | SS316L | 586.25 | 2.67 | 626.97 | 2.85 | 630.89 | 2.87 | 430.00 | 1.96 | 417.80 | 1.90 | 473.22 | 2.15 | 476.05 | 2.16 | 340.19 | 1.55 | 568.57 | 2.58 | 645.50 | 2.93 | 630.53 | 2.87 | 431.92 | 1.96 |
| | CoCrMo | 586.24 | 2.79 | 626.96 | 2.99 | 630.88 | 3.01 | 429.99 | 2.05 | 417.79 | 1.99 | 473.20 | 2.25 | 476.04 | 2.27 | 340.19 | 1.62 | 568.56 | 2.71 | 645.49 | 3.07 | 630.52 | 3.00 | 431.92 | 2.06 |
| | Ti6Al4V | 586.09 | 5.33 | 626.77 | 5.70 | 630.71 | 5.74 | 429.90 | 3.91 | 417.66 | 3.80 | 472.95 | 4.30 | 475.83 | 4.33 | 340.08 | 3.10 | 568.44 | 5.17 | 645.35 | 5.87 | 630.38 | 5.73 | 431.84 | 3.93 |
| M2 | SS316L | 375.40 | 1.71 | 398.86 | 1.81 | 398.76 | 1.81 | 284.69 | 1.29 | 290.95 | 1.32 | 327.37 | 1.49 | 325.43 | 1.48 | 238.28 | 1.08 | 360.39 | 1.64 | 400.86 | 1.82 | 388.20 | 1.76 | 282.07 | 1.28 |
| | CoCrMo | 375.40 | 1.79 | 398.86 | 1.90 | 398.76 | 1.90 | 284.69 | 1.36 | 290.95 | 1.39 | 327.37 | 1.56 | 325.43 | 1.55 | 238.28 | 1.13 | 360.39 | 1.72 | 400.86 | 1.91 | 388.20 | 1.85 | 282.07 | 1.34 |
| | Ti6Al4V | 375.40 | 3.41 | 398.86 | 3.63 | 398.76 | 3.63 | 284.69 | 2.59 | 290.95 | 2.65 | 327.37 | 2.98 | 325.43 | 2.96 | 238.28 | 2.17 | 360.39 | 3.28 | 400.86 | 3.64 | 388.20 | 3.53 | 282.07 | 2.56 |
| M3 | SS316L | 193.05 | 0.88 | 203.20 | 0.92 | 200.82 | 0.91 | 155.19 | 0.71 | 194.50 | 0.88 | 215.74 | 0.98 | 214.36 | 0.97 | 160.93 | 0.73 | 185.94 | 0.85 | 202.25 | 0.92 | 197.09 | 0.90 | 153.89 | 0.70 |
| | CoCrMo | 193.05 | 0.92 | 203.20 | 0.97 | 200.82 | 0.96 | 155.19 | 0.74 | 194.50 | 0.93 | 215.74 | 1.03 | 214.36 | 1.03 | 160.93 | 0.77 | 185.94 | 0.89 | 202.25 | 0.96 | 197.09 | 0.94 | 153.89 | 0.73 |
| | Ti6Al4V | 193.05 | 1.76 | 203.20 | 1.85 | 200.82 | 1.83 | 155.19 | 1.41 | 194.50 | 1.77 | 215.74 | 1.96 | 214.36 | 1.95 | 160.93 | 1.46 | 185.94 | 1.69 | 202.25 | 1.84 | 197.09 | 1.79 | 153.89 | 1.40 |
| M4 | SS316L | 144.55 | 0.66 | 151.49 | 0.69 | 148.11 | 0.67 | 118.04 | 0.54 | 157.15 | 0.71 | 173.81 | 0.79 | 172.06 | 0.78 | 131.86 | 0.60 | 134.20 | 0.61 | 145.65 | 0.66 | 140.77 | 0.64 | 114.30 | 0.52 |
| | CoCrMo | 144.55 | 0.69 | 151.49 | 0.72 | 148.11 | 0.71 | 118.04 | 0.56 | 157.15 | 0.75 | 173.81 | 0.83 | 172.06 | 0.82 | 131.86 | 0.63 | 134.20 | 0.64 | 145.65 | 0.69 | 140.77 | 0.67 | 114.30 | 0.54 |
| | Ti6Al4V | 144.55 | 1.31 | 151.49 | 1.38 | 148.11 | 1.35 | 118.04 | 1.07 | 157.15 | 1.43 | 173.81 | 1.58 | 172.06 | 1.56 | 131.86 | 1.20 | 134.20 | 1.22 | 145.65 | 1.32 | 140.77 | 1.28 | 114.30 | 01.04 |

**Table A2.** *Cont.*

| Location | Material | P1-Circular σ (MPa) | ε (mm/m) | P1-Ellipse σ (MPa) | ε (mm/m) | P1-Oval σ (MPa) | ε (mm/m) | P1-Trapezoidal σ (MPa) | ε (mm/m) | P2-Circular σ (MPa) | ε (mm/m) | P2-Ellipse σ (MPa) | ε (mm/m) | P2-Oval σ (MPa) | ε (mm/m) | P2-Trapezoidal σ (MPa) | ε (mm/m) | P3-Circular σ (MPa) | ε (mm/m) | P3-Ellipse σ (MPa) | ε (mm/m) | P3-Oval σ (MPa) | ε (mm/m) | P3-Trapezoidal σ (MPa) | ε (mm/m) |
|---|---|---|---|---|---|---|---|---|---|---|---|---|---|---|---|---|---|---|---|---|---|---|---|---|---|
| M5 | SS316L | 109.54 | 0.50 | 112.27 | 0.51 | 113.46 | 0.52 | 90.34 | 0.41 | 122.15 | 0.56 | 134.58 | 0.61 | 133.10 | 0.61 | 104.58 | 0.48 | 106.25 | 0.48 | 115.06 | 0.52 | 113.08 | 0.51 | 92.55 | 0.42 |
| | CoCrMo | 109.54 | 0.52 | 112.27 | 0.54 | 113.46 | 0.54 | 90.34 | 0.43 | 122.15 | 0.58 | 134.58 | 0.64 | 133.10 | 0.64 | 104.58 | 0.50 | 106.25 | 0.51 | 115.06 | 0.55 | 113.08 | 0.54 | 92.55 | 0.44 |
| | Ti6Al4V | 109.54 | 1.00 | 112.27 | 1.02 | 113.46 | 1.03 | 90.34 | 0.82 | 122.15 | 1.11 | 134.58 | 1.23 | 133.10 | 1.21 | 104.58 | 0.95 | 106.25 | 0.97 | 115.06 | 1.05 | 113.08 | 1.03 | 92.55 | 0.84 |
| M6 | SS316L | 173.89 | 0.79 | 179.77 | 0.82 | 179.70 | 0.82 | 145.28 | 0.66 | 167.54 | 0.76 | 187.28 | 0.85 | 186.55 | 0.85 | 150.12 | 0.68 | 167.81 | 0.76 | 182.89 | 0.83 | 185.76 | 0.84 | 150.09 | 0.68 |
| | CoCrMo | 173.89 | 0.83 | 179.77 | 0.86 | 179.70 | 0.86 | 145.28 | 0.69 | 167.54 | 0.80 | 187.28 | 0.89 | 186.55 | 0.89 | 150.12 | 0.72 | 167.81 | 0.80 | 182.89 | 0.87 | 185.76 | 0.88 | 150.09 | 0.72 |
| | Ti6Al4V | 173.89 | 1.58 | 179.77 | 1.64 | 179.70 | 1.63 | 145.28 | 1.32 | 167.54 | 1.52 | 187.28 | 1.70 | 186.55 | 1.70 | 150.12 | 1.37 | 167.81 | 1.53 | 182.89 | 1.66 | 185.76 | 1.69 | 150.09 | 1.37 |
| M7 | SS316L | 237.88 | 1.08 | 242.93 | 1.10 | 245.57 | 1.12 | 205.28 | 0.93 | 221.40 | 1.01 | 253.48 | 1.15 | 258.50 | 1.18 | 211.66 | 0.96 | 230.22 | 1.05 | 263.50 | 1.20 | 265.28 | 1.21 | 219.21 | 1.00 |
| | CoCrMo | 237.88 | 1.13 | 242.93 | 1.16 | 245.57 | 1.17 | 205.28 | 0.98 | 221.40 | 1.05 | 253.48 | 1.21 | 258.50 | 1.23 | 211.66 | 1.01 | 230.22 | 1.10 | 263.50 | 1.26 | 265.28 | 1.26 | 219.21 | 1.05 |
| | Ti6Al4V | 237.88 | 2.16 | 242.93 | 2.21 | 245.57 | 2.23 | 205.28 | 1.87 | 221.40 | 2.01 | 253.48 | 2.31 | 258.50 | 2.35 | 211.66 | 1.93 | 230.22 | 2.09 | 263.51 | 2.40 | 265.28 | 2.41 | 219.21 | 2.00 |
| M8 | SS316L | 295.92 | 1.35 | 297.82 | 1.35 | 302.41 | 1.38 | 268.88 | 1.22 | 271.10 | 1.23 | 314.21 | 1.43 | 321.58 | 1.46 | 279.16 | 1.27 | 295.93 | 1.35 | 338.91 | 1.54 | 348.42 | 1.58 | 305.08 | 1.39 |
| | CoCrMo | 295.92 | 1.41 | 297.82 | 1.42 | 302.41 | 1.44 | 268.88 | 1.28 | 271.10 | 1.29 | 314.21 | 1.50 | 321.58 | 1.53 | 279.16 | 1.33 | 295.93 | 1.41 | 338.91 | 1.61 | 348.42 | 1.66 | 305.08 | 1.46 |
| | Ti6Al4V | 295.92 | 2.69 | 297.81 | 2.71 | 302.40 | 2.75 | 268.88 | 2.45 | 271.11 | 2.47 | 314.23 | 2.86 | 321.60 | 2.92 | 279.19 | 2.54 | 295.93 | 2.69 | 338.93 | 3.08 | 348.45 | 3.17 | 305.11 | 2.78 |
| L1 | SS316L | 394.06 | 1.79 | 424.21 | 1.93 | 399.38 | 1.82 | 253.62 | 1.15 | 266.40 | 1.21 | 310.60 | 1.41 | 285.55 | 1.30 | 189.18 | 0.86 | 374.30 | 1.70 | 426.20 | 1.94 | 393.86 | 1.79 | 251.09 | 1.14 |
| | CoCrMo | 394.06 | 1.88 | 424.21 | 2.02 | 399.38 | 1.90 | 253.62 | 1.21 | 266.40 | 1.27 | 310.60 | 1.48 | 285.55 | 1.36 | 189.18 | 0.90 | 374.30 | 1.78 | 426.20 | 2.03 | 393.86 | 1.88 | 251.10 | 1.20 |
| | Ti6Al4V | 394.06 | 3.58 | 424.21 | 3.86 | 399.38 | 3.63 | 253.64 | 2.31 | 266.41 | 2.42 | 310.60 | 2.82 | 285.57 | 2.60 | 189.22 | 1.72 | 374.30 | 3.40 | 426.20 | 3.87 | 393.86 | 3.58 | 251.12 | 2.28 |
| L2 | SS316L | 198.81 | 0.90 | 214.57 | 0.98 | 199.13 | 0.91 | 136.60 | 0.62 | 175.63 | 0.80 | 200.49 | 0.91 | 186.50 | 0.85 | 125.65 | 0.57 | 190.88 | 0.87 | 209.72 | 0.95 | 194.72 | 0.89 | 134.50 | 0.61 |
| | CoCrMo | 198.81 | 0.95 | 214.57 | 1.02 | 199.13 | 0.95 | 136.60 | 0.65 | 175.63 | 0.84 | 200.49 | 0.95 | 186.50 | 0.89 | 125.65 | 0.60 | 190.88 | 0.91 | 209.72 | 1.00 | 194.72 | 0.93 | 134.50 | 0.64 |
| | Ti6Al4V | 198.81 | 1.81 | 214.57 | 1.95 | 199.13 | 1.81 | 136.60 | 1.24 | 175.63 | 1.60 | 200.49 | 1.82 | 186.50 | 1.70 | 125.65 | 1.14 | 190.88 | 1.74 | 209.72 | 1.91 | 194.72 | 1.77 | 134.50 | 1.22 |
| L3 | SS316L | 57.88 | 0.26 | 61.82 | 0.28 | 55.60 | 0.25 | 41.82 | 0.19 | 109.31 | 0.50 | 123.03 | 0.56 | 114.91 | 0.52 | 79.07 | 0.36 | 69.63 | 0.32 | 75.81 | 0.34 | 69.64 | 0.32 | 52.36 | 0.24 |
| | CoCrMo | 57.88 | 0.28 | 61.82 | 0.29 | 55.60 | 0.27 | 41.82 | 0.20 | 109.31 | 0.52 | 123.03 | 0.59 | 114.91 | 0.55 | 79.07 | 0.38 | 69.63 | 0.33 | 75.81 | 0.36 | 69.64 | 0.33 | 52.36 | 0.25 |
| | Ti6Al4V | 57.88 | 0.53 | 61.82 | 0.56 | 55.60 | 0.51 | 41.82 | 0.38 | 109.31 | 0.99 | 123.03 | 1.12 | 114.91 | 1.04 | 79.07 | 0.72 | 69.63 | 0.63 | 75.81 | 0.69 | 69.64 | 0.63 | 52.36 | 0.48 |
| L4 | SS316L | 48.61 | 0.22 | 51.18 | 0.23 | 47.00 | 0.21 | 34.26 | 0.16 | 75.88 | 0.35 | 85.54 | 0.39 | 78.72 | 0.36 | 55.14 | 0.25 | 18.70 | 0.086 | 21.02 | 0.10 | 18.27 | 0.08 | 13.74 | 0.06 |
| | CoCrMo | 48.61 | 0.23 | 51.18 | 0.24 | 47.00 | 0.22 | 34.26 | 0.16 | 75.88 | 0.36 | 85.54 | 0.41 | 78.72 | 0.38 | 55.14 | 0.26 | 18.70 | 0.09 | 21.02 | 0.10 | 18.27 | 0.09 | 13.74 | 0.07 |
| | Ti6Al4V | 48.61 | 0.44 | 51.18 | 0.47 | 47.00 | 0.43 | 34.26 | 0.31 | 75.88 | 0.69 | 85.54 | 0.78 | 78.72 | 0.72 | 55.14 | 0.50 | 18.70 | 0.17 | 21.02 | 0.19 | 18.27 | 0.17 | 13.74 | 0.13 |
| L5 | SS316L | 37.50 | 0.17 | 39.93 | 0.18 | 39.74 | 0.18 | 28.79 | 0.13 | 15.92 | 0.08 | 17.08 | 0.08 | 17.28 | 0.08 | 12.22 | 0.06 | 64.10 | 0.30 | 72.64 | 0.34 | 66.30 | 0.31 | 51.97 | 0.24 |
| | CoCrMo | 37.50 | 0.18 | 39.93 | 0.19 | 39.74 | 0.19 | 28.79 | 0.14 | 15.92 | 0.08 | 17.08 | 0.08 | 17.28 | 0.08 | 12.22 | 0.06 | 64.10 | 0.31 | 72.64 | 0.35 | 66.30 | 0.32 | 51.97 | 0.25 |
| | Ti6Al4V | 37.50 | 0.34 | 39.93 | 0.37 | 39.74 | 0.36 | 28.79 | 0.26 | 15.92 | 0.16 | 17.08 | 0.16 | 17.28 | 0.16 | 12.22 | 0.11 | 64.10 | 0.59 | 72.64 | 0.67 | 66.30 | 0.61 | 51.97 | 0.48 |
| L6 | SS316L | 126.67 | 0.58 | 128.48 | 0.58 | 122.87 | 0.56 | 92.73 | 0.42 | 105.73 | 0.48 | 120.82 | 0.55 | 113.52 | 0.52 | 86.33 | 0.39 | 135.05 | 0.61 | 148.40 | 0.68 | 141.91 | 0.65 | 106.22 | 0.48 |
| | CoCrMo | 126.67 | 0.60 | 128.48 | 0.61 | 122.87 | 0.59 | 92.73 | 0.44 | 105.73 | 0.50 | 120.82 | 0.58 | 113.52 | 0.54 | 86.33 | 0.41 | 135.05 | 0.64 | 148.40 | 0.71 | 141.91 | 0.68 | 106.22 | 0.51 |
| | Ti6Al4V | 126.67 | 1.15 | 128.48 | 1.17 | 122.87 | 1.12 | 92.73 | 0.84 | 105.73 | 0.96 | 120.82 | 1.10 | 113.52 | 1.03 | 86.33 | 0.79 | 135.05 | 1.23 | 148.40 | 1.35 | 141.91 | 1.29 | 106.22 | 0.97 |
| L7 | SS316L | 185.91 | 0.85 | 189.94 | 0.86 | 181.03 | 0.82 | 141.37 | 0.64 | 166.70 | 0.76 | 191.35 | 0.87 | 185.52 | 0.84 | 143.83 | 0.65 | 188.30 | 0.86 | 212.52 | 0.97 | 204.10 | 0.93 | 159.90 | 0.73 |
| | CoCrMo | 185.91 | 0.89 | 189.94 | 0.91 | 181.03 | 0.86 | 141.37 | 0.67 | 166.70 | 0.79 | 191.35 | 0.91 | 185.52 | 0.88 | 143.82 | 0.69 | 188.30 | 0.90 | 212.52 | 1.01 | 204.10 | 0.97 | 159.90 | 0.76 |
| | Ti6Al4V | 185.91 | 1.69 | 189.94 | 1.73 | 181.03 | 1.65 | 141.37 | 1.29 | 166.70 | 1.52 | 191.35 | 1.74 | 185.52 | 1.69 | 143.82 | 1.31 | 188.30 | 1.71 | 212.52 | 1.93 | 204.10 | 1.86 | 159.90 | 1.46 |
| L8 | SS316L | 243.01 | 1.11 | 248.30 | 1.13 | 241.06 | 1.10 | 206.98 | 0.94 | 218.45 | 0.99 | 261.10 | 1.19 | 253.35 | 1.15 | 213.86 | 0.97 | 246.25 | 1.12 | 288.76 | 1.31 | 281.91 | 1.28 | 244.16 | 1.11 |
| | CoCrMo | 243.01 | 1.16 | 248.30 | 1.18 | 241.06 | 1.15 | 206.98 | 0.99 | 218.45 | 1.04 | 261.10 | 1.24 | 253.35 | 1.21 | 213.86 | 1.02 | 246.25 | 1.17 | 288.76 | 1.38 | 281.91 | 1.34 | 244.16 | 1.16 |
| | Ti6Al4V | 243.00 | 2.21 | 248.30 | 2.26 | 241.07 | 2.19 | 206.98 | 1.88 | 218.44 | 1.99 | 261.08 | 2.37 | 253.34 | 2.30 | 213.83 | 1.95 | 246.25 | 2.24 | 288.73 | 2.63 | 281.87 | 2.57 | 244.13 | 2.22 |

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
