# Peer review of "Finite Element Assessment of a Hybrid Proposal for Hip Stem, from a Standardized Base and Different Activities"

_applsci, doi:10.3390/app12167963_

Round 1

Reviewer 1 Report

This work aims to use a standardized base to compare the most common stem geometries found in the literature and create a new hybrid design whose performance improves from the commercially available, so it will be possible to obtain and compare their best qualities under different activities. Sixteen different stems identified from the literature review were analyzed by using multi-objective under the same boundary and load conditions. As a result, the von Misses stress and strain were obtained. The results were validated by the Finite Element Method (FEM) and compared with previous works and those with the ISO standard. 

Before the Editor makes a decision, I suggest that the authors must take into account the following corrections:

1.      The title should be rewritten to be friendly. I suggest “Finite element assessment of a hybrid proposal for hip stem, from a standardized base and different activities.

2.      The author should explain the novelty clearly in the abstract and conclusion.

3.      The abstract is very long, should be reduced

4.      The Introduction section must be more concise.

5.      The paper should be carefully revised for punctuation, grammar, spelling mistakes and sentences structuring.

6.      What are the advantages of the proposed new model?

7.      The model has been solved numerically but authors did not provided any information regarding their methodology such as grid size, time step, order of accuracy of the numerical scheme etc.

10.   What the type of element?

11.   What the number of element?

12.   The literature survey might be improved on adding some relevant references as:

·       "A direct finite element method study of generalized thermoelastic problems." International Journal of Solids and Structures 43.7-8 (2006): 2050-2063.

·       LS model on thermal shock problem of generalized magneto-thermoelasticity for an infinitely long annular cylinder with variable thermal conductivity." Applied Mathematical Modelling 35.8 (2011): 3759-3768.

·       "Implementation of the extended finite element method for dynamic thermoelastic fracture initiation." International Journal of Solids and Structures 47.10 (2010): 1392-1404.

·       Nonlinear transient thermal stress analysis of thick-walled FGM cylinder with temperature-dependent material properties. Meccanica, 2014. 49(7): p. 1697-1708.

·       "Thermoelastic deflection responses of CNT reinforced sandwich shell structure using finite element method." Scientia Iranica 25.5 (2018): 2722-2737.

If the authors take into account all these corrections, then, without doubt, this manuscript deserves to be published. 

Author Response

Dear reviewer, first and foremost, thanks for sharing your comments and experience with us, we have attended to all your comments, and the answers are attached.

Reviewer 2 Report

Thank you very much for inviting me to review the manuscript "Finite element assessment of a hybrid proposal hip stem, from a standardized base and different activities” The manuscript is interesting, but from my humble point of view, this requires some changes and explanations that should be made previously that the manuscript will be published in Applied Sciences 

 Major Corrections:  

  • The introduction needs to be improved. At the end of the introduction, there should be a paragraph showing clearly what is going to be developed in the manuscript. Please, improve this paragraph at the end of the introduction. 
  • Has any type of algorithm been proposed for the multi-objective analysis? What type of algorithm was used in this case? Was any design of experiments carried out in this work to study the different geometries proposed? this point should be explained in detail as it is very important and is not clear in the manuscript. 
  • What type of elements has been considered in this study? Please justify.    
  • What type of formulation has been adopted for the elements selected? Please specify.   
  • Has any mesh quality analysis been carried out on the proposed FE model (aspect ratio, Jacobian rate, etc.)? Please justify.   
  • The boundary conditions of the FE proposed should be better explained. Please improve the description of this section.  

In order to facilitate the authors to improve their manuscript, the following papers show how their authors solved some of the key questions and points that should be improved in this manuscript (Mesh sensitivity analysis, Boundary condictions, multi-objective analysis, etc). I suggest that the authors review the attached papers as they can facilitate the improvement of their manuscript.   

  • Comparative Analysis of Healthy and Cam-Type Femoroacetabular Impingement (FAI) Human Hip Joints Using the Finite Element Method  
  • A proposed methodology for setting the finite element models based on healthy human intervertebral lumbar discs 
  • Finite element model updating combined with multi-response optimization for hyper-elastic materials characterization 
  • Improving the process of adjusting the parameters of finite element models of healthy human intervertebral discs by the multi-response surface method   
  • Improvement in determining the risk of damage to the human lumbar functional spinal unit considering age, height, weight and sex using a combination of FEM and RSM  
  • Improvement in the process of designing a new artificial human intervertebral lumbar disc combining soft computing techniques and the finite element method  

Minor Corrections:   

  • The authors must be sure that the figures, tables and references must be in accordance with the format of the journal. 

Author Response

Dear reviewer, first and foremost, thanks for sharing your comments and experience with us, we have attended to all your comments, and the answers are attached.

B.R

The team

Reviewer 3 Report

The authors aimed to develop a hybrid design based on the best mechanical performance of the previous implants studied, resulting in lower stresses and strains in the  analyzed points.

The data were important, because designing a new prosthesis by doing a geometry modification or using an optimal biomaterial can be a solution to reduce potential problems like aseptic loosening or a fracture of the implant increase due to the lack of edges that increase the contact area.

Therefore, I have no further comments against the manuscript.

Author Response

Dear reviewer, first and foremost, thanks for sharing your comments and experience with us. We are enthusiastic about this research.

B.W.

The team

Round 2

Reviewer 2 Report

After analyzing the review carried out by the authors, as well as the response they have given to the rest of the reviewers, from my point of view the article can definitely be published in the journal: Applied of Science.